# Comparing Offshore Ferry Lidar Measurements in the Southern Baltic Sea with ASCAT, FINO2 and WRF

**Daniel Hatfield** [1,*]🆔**, Charlotte Bay Hasager** [1]🆔 **and Ioanna Karagali** [2]🆔

1 Department of Wind Energy, Technical University of Denmark, Frederiksborgvej 399, 4000 Roskilde, Denmark; cbha@dtu.dk
2 Danmarks Meteorologiske Institut, Lyngbyvej 100, 2100 Copenhagen, Denmark; ika@dmi.dk
* Correspondence: dhat@dtu.dk; Tel.: +45-93510512

**Abstract:** This article highlights the inter-comparisons of the wind measurement techniques available in deep water areas working towards combining them to obtain optimal estimates of the wind power potential. More specifically, this article presents comparisons of the Ferry Lidar Experiment wind data with those of the Advanced Scatterometer (ASCAT), the FINO2 meteorological mast, and the New European Wind Atlas (NEWA) simulations performed using the Weather Research, and Forecasting (WRF) mesoscale model. To be comparable to ASCAT surface winds, which are referenced at 10 m, the ferry lidar and FINO2 wind profile measurements were extrapolated down to 10 m using atmospheric stability information derived from the bulk Richardson number formulation. ASCAT had the lowest associated error compared with that of the ferry lidar in near-neutral atmospheric stratifications, whereas FINO2, despite a distance range of 30 km and a moving ferry lidar target, had the highest correlation and lowest RMSE in all atmospheric conditions. Due to the high frequency of low-level jets caused by the proximity to land from all directions as well as typically stable atmospheric conditions, the extrapolated ferry lidar measurements underpredicted the ASCAT 10 m wind speeds. WRF consistently underperformed compared to the other measurement methods, even with the ability to directly compare results with all other sources at all heights.

**Keywords:** offshore wind energy; ASCAT; satellite winds; wind lidar; ferry lidar; WRF; FINO2; NEWA

## 1. Introduction

Offshore wind energy capacity is expected to grow from 12 GW in 2020 to 30 GW in 2030 [1]. Since 1991, when the first offshore wind turbine was installed at Vindeby, Denmark [2], the scale and demand for offshore wind is at its highest. With the increasing size of offshore wind farms, the average water depth for installation has increased, furthering the need for more robust planning and development. Subsequently, a solid basis of wind information is necessary to estimate the future energy yield, expected return, and design optimization of the planned investment. Offshore wind resource has become an area of interest, with increasing installation of wind farms in coastal areas and with the move to further depths (>50 m) such as the Hywind project [3]. Traditional ground-fixed meteorological masts (met masts) have been used offshore to study the wind climate near wind farms such as the German Forschungsplattformen In Nord-und Ostsee (FINO) masts in the North and Baltic Seas [4]; however, when considering far offshore locations it becomes extremely costly and complicated to install, so remote sensing alternatives are favored.

The main alternative to offshore met masts have been lidar systems [5]; more specifically, floating lidar systems (FLS) [6]. FLS research has thus increased dramatically during the last few years with the first buoy-lidar system installed and tested in 2009 [7]. The superior flexibility and lower cost compared to the traditional met mast is an incentive; however, the reliability needs to be addressed before it becomes an industrial standard [8]. The first ship-based lidar measurements were performed by Fraunhofer IWES in [9], where the

motion correction algorithm was adapted for ship-based floating systems from [10] while being tested in [9,11]. The sea-induced movement correction is dependent on the FLS itself—either employing motion compensation (i.e., from a stability platform) or motion detection using suitable sensors. The verification of FLS motion correction is typically studied in two ways: comparison with mesoscale models [12] or in situ data [9,13]. This means that for ship-borne lidar in deep-water locales, the main validation technique at multiple heights, so far, has been comparison with mesoscale models.

The New European Wind Atlas (NEWA) [14] provides a publicly available wind resource dataset for Europe. It is based on 30 years of mesoscale simulations using the Weather Research and Forecasting (WRF) model [15], available every 30 min with a 3 km × 3 km grid resolution. As mesoscale models are not specifically developed for wind energy applications, the main objective for the NEWA project was to gather the best practices and to create a unified modeling methodology [14]. Although WRF has been shown to compare well with measurements at relatively shallow-water offshore locations [16], deep sea sites have not been assessed, and site-specific measurements are still vital to study the wind climate.

Satellite wind retrievals can provide observations of the ocean surface at improved spatial resolution compared to those of mesoscale models [17]. Surface-derived winds from scatterometer instruments such as the Advanced Scatterometer (ASCAT) are obtained at a height of 10 m and have been used for offshore wind energy application studies [18]. While not directly applicable to wind observations needed for wind farm hub height measurements (100+ m), they do provide daily global wind field measurements at the ocean surface. Vertical extrapolation of satellite data has been performed by [19–21], bringing surface winds to hub heights using the long-term stability correction from [22]. These require longer time scales and accurate stability information, whereas vertically extrapolating co-located ASCAT points to the moving instantaneous wind profiles of the ferry lidar adds another level of complexity. However, extrapolating lidar or offshore met masts profiles using surface layer theory down to the 10 m height in order to make a direct comparison with ASCAT is possible. The same process can be performed with the nearby German meteorological mast, FINO2, with a much larger, stationary dataset that can be used as standard for comparison.

This region of the Baltic Sea was studied extensively in [23] in the different methodologies of offshore wind resource assessment and in [24,25] for low-level jet activity. Ref. [26] looked at mesoscale processes over the area, incorporating lidar measurements. Refs. [20,27,28] studied satellite synthetic aperture radar (SAR) and ASCAT measurements over the area of interest with [19], using the FINO2 location as reference for the long-term stability correction used in profile extrapolation. A study of the NEWA WRF model production runs used in this study with that of the FINO met masts is detailed and presented in [14].

The purpose of this work is to assess the strengths and limitations of the aforementioned ferry lidar experiment using the available datasets from ASCAT and FINO2 in the remote offshore locations of the southern Baltic Sea. This study also highlights strengths and weaknesses of the various sources of data, working towards combining them to obtain optimal estimates of the wind power potential. These techniques alongside mesoscale model simulations such as the NEWA WRF dataset can then be used to study potential regions for entirely floating wind farms in deep water locales.

The various datasets and filtering processes used throughout this study are described in Section 2, as well as the methods of data co-location and stability calculations. Section 3 presents the results of the ferry lidar with ASCAT, FINO2, and WRF as well as an intercomparison of the previously mentioned datasets. This section also introduces the low-level jet results from the ferry lidar experiment. Section 4 discusses the advantages and disadvantages of using a ferry-based compared to buoy-based lidar, as well as difficulties of interpreting results from ferry lidar measurements as compared with other data sources investigated in this work. Finally, conclusions and outlook are outlined in Section 5.

## 2. Measurements and Data Processing

The interest area of the southern Baltic Sea spans from 53°N to 56°N and from 10°E to 21°E with an average water depth of 55 m. There are 11 wind farms located within the aforementioned area of interest, with four having potential influence on the wind in the path of the ferry lidar (Nysted, Rødsand II, EnBW Baltic I & II).

In this study, multiple data sources were used; NEWA ferry lidar wind measurements and meteorological sensor data, Operational Sea Surface Temperature and Ice Analysis (OSTIA) daily satellite sea-surface temperature (SST) data, FINO2 wind and meteorological data, ASCAT wind product data, and NEWA WRF production-run mesoscale model data. These datasets will be explained in more detailed with the time period in which they were used in the following sections. All datasets were recorded in Coordinated Universal Time (UTC).

### 2.1. Kiel Ferry Lidar

The NEWA ferry lidar campaign was conducted in two phases: the first one as a preparatory campaign in the North Sea from Bremerhaven to Helgoland [29] for 2 months in 2016. The main campaign was a four-month campaign from 7 February 2017 to 6 June 2017 in the Baltic Sea from Kiel, Germany to Klaipeda, Lithuania outlined in [12] and was the only offshore measurement campaign in the NEWA project.

A WindCube v2 lidar was installed on top of the vessel Victoria Seaways, which belongs to the DFDS Seaways Group, where it ran almost continuously (20 h at sea, 4 h in the harbor). Due to the length of the trip, measurements were recorded on average in the same location every second day with small deviations occurring further away from the harbor. Wind measurements were performed at twelve different altitudes from 65 m to 275 m above sea level (accounting for the 25 m to the deck height). Meteorological parameters (temperature, humidity, air pressure, and precipitation) were also recorded along with a weather station, motion sensors, and a satellite compass installed near the lidar position.

The processed dataset used comprised motion-corrected horizontal wind speed and directions, carrier-to-noise ratios (CNR), and ship positions and time stamps sampled at 0.7 s. Similar to the original campaign, CNR values below $-29$ dB were omitted as per the recommendation from the manufacturer. Data availability for 10 min averages in the four-month period was well above 90% below 150 m, with less availability at the higher heights down to 40% at 275 m. See [12] for the full data report.

### 2.2. Sea Surface Temperature

The ship lidar system had no sea surface temperature (SST) system aside from the ship SST measurements which were deemed untrustworthy [12], so daily satellite SST data were used from Global Ocean OSTIA Sea Surface Temperature for the Baltic Sea area. This product has a spatial resolution of 1/20° and a temporal resolution of 1 day. OSTIA SST has zero mean bias and an accuracy of 0.57 K compared to in situ measurements [30], and our comparison with the FINO2 SST at 2 m data yielded a correlation of $R^2 = 0.99$ (not shown).

### 2.3. FINO2

The FINO project began in the early 2000s and now consists of two offshore research platforms in the North Sea and one in the Baltic Sea, all of which have 100 m meteorological towers (see [4]). Meteorological parameters are recorded at frequencies of 1–10 Hz, and data are averaged in intervals ranging from 10 to 30 min. FINO2 is the met mast in the Baltic Sea located at 55.0069°N–13.1542°E, situated 33 km north of the German island Rügen between the borders of Denmark, Germany, and Sweden. FINO2 is installed three kilometers north of the EnBW Baltic 2 wind farm, where the water depth is 35 m. The observations used were taken from the period where SST measurements began, 1 April 2013, to 30 November 2017. The relevant quantities were wind speeds at heights (32 m, 42 m, 52 m, 62 m, 72 m, 82 m, 92 m, 102 m) and wind directions at (31 m, 51 m, 71 m, 91 m) as well as meteorological

data of air pressure (30 m), air temperature (30 m), sea surface temperature (2 m depth), and relative humidity (30 m).

The minimum distance of the ferry lidar track to FINO2 is 24 km; therefore, a range of 30 km was used to have a suitable number of collocated observations to directly compare wind speeds at 62 m, 72 m, 92, and 102 m, as well as wind directions at 71 m and 91 m. The collocations occurred between 07:00–08:00 during the eastward path of the ferry and between 23:00–00:00 on the westward path. Due to the separation distance, different time scales were used for the comparison at 10 min, 30 min, and 1 h, where the 1 h timescale consistently yielded the best results. Different wind speed thresholds, time lags, wind direction dependencies, and distance ranges were tested. The 1 h direct comparison showed the most stable performance.

### 2.4. ASCAT

The Advanced Scatterometer is an instrument launched by the European Space Agency (ESA) flown on the Meteorological Operational (Metop) satellites developed by the European Organization for the Exploitation of Meteorological Satellites (EUMETSAT) [31]. It was first launched on Metop-A in October 2006, launched as payload on Metop-B in September 2012, and also on Metop-C in November 2018. The instrument has an effective swath width of 512.5 km with a nadir gap of 700 km, leading to 1–3 passes daily from both the ascending and descending passes, depending on the time period.

The EUMETSAT ASCAT wind products provide wind speed and direction measurements at 10 m above the sea surface. The data used in this study have a grid spacing of 12.5 km and a spatial resolution of 25 km [31]. This dataset is processed and distributed by EUMETSAT Ocean and Sea Ice (OSI) Satellite Application facility (SAF) as well as Advanced Retransmission Service (EARS) ground system. Both of these are implemented at the Koninklijk Nederlands Meteorologisch Instituut (KNMI) and are freely available worldwide (see knmi.nl). For the present study, data were obtained through the Copernicus Marine Service (last accessed 25 February 2022) https://resources.marine.copernicus.eu/, using product WIND_GLO_WIND_L3_NRT_OBSERVATIONS_012_002 from 2016 and onwards, while from 2007–2015 product WIND_GLO_WIND_L3_REP_OBSERVATIONS_012_005 was used.

A scatterometer is an active radar that measures the backscatter power from transmitted pulses. This backscatter depends on the surface roughness; a completely smooth surface will result in reflection with no backscatter back to the instrument. As the roughness increases, the backscatter component increases simultaneously [32]. The wind stress creates small-scale (cm) capillary waves resulting from the surface tension. Large-scale waves or tidal influences have lower resulting backscatter influence due to the wavelength (C band). As the wind speed increases, so too does the surface tension and therefore the amplitude of the resulting capillary waves. In effect, this increases the surface roughness and thereby allows for more backscatter. The measurements of the backscatter power from the observed area can be used to estimate the normalized radar cross section (NRCS, $\sigma_0$) [33]. The NRCS is the relation between the received and transmitted power which depends on the radar settings, the atmospheric attenuation, and the ocean surface characteristics [34]. The computation of ocean surface winds from the NRCS comes from an empirically derived geophysical model function (GMF) [35].

The physical definition of the ASCAT winds are horizontal stress-equivalent (SE) winds obtained using the CMOD7 GMF [35,36]. SE winds are estimated from the sea surface roughness to the 10 m height independent of atmospheric conditions. In order to obtain SE winds from measured 10 m winds, one must first compute the equivalent neutral winds at 10 m and then multiply by a density correction factor. Since the atmospheric boundary is, on average, weakly unstable, the SE wind will be, on average, slightly stronger than real wind by approximately 0.1 to 0.2 m s$^{-1}$, with a standard deviation of 1.7 m s$^{-1}$ for the 12.5 km product and a bias of 0.02 m s$^{-1}$ [31].

### 2.5. Collocation Procedure

During the four-month ferry lidar campaign, 126 wind vector cell pairs (WVCs) from ASCAT were collocated with the lidar profile near its center, where 119 had a fully available wind profile. For any given collocation, the ferry lidar wind profiles were then averaged about the central collocation point for a total period of 1 h which, with the ship's speed and direction, spatially covered the two ASCAT cells in the ship's path (see Figure 1). Note that some of the ferry lidar averaged data will fall outside of the ASCAT grid cells in the open sea where the ship had a higher overall speed compared to locations closer to the two harbors. The position of each collocated central lidar profile on the ferry is shown in Figure 2. Due to the sun-synchronous orbit of the satellite and the ferry schedule, the points clustered closer to shore during the nighttime (typically between 19:00–21:00) in the western part of the transect, whereas the points in the open water near FINO2 and outside of the harbor in Klaipeda were measured during the day (between 8:00–11:00) in the eastern part of the transect.

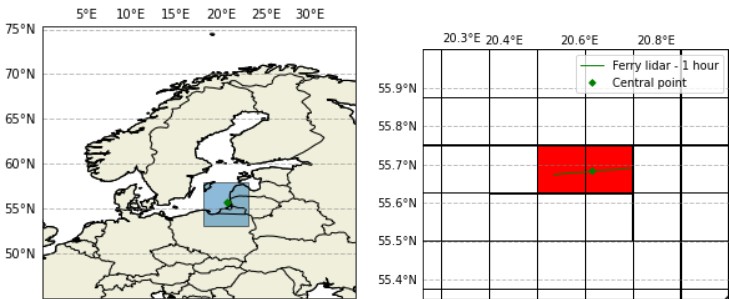

**Figure 1. Left**: Position of point in Europe. **Right**: ASCAT grid cell pair and corresponding 1 h of the ferry lidar path on 2 September 2017.

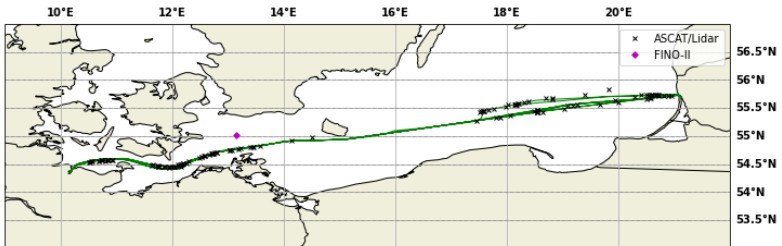

**Figure 2.** Collocation of ASCAT grid cells with Kiel ferry lidar path with Kiel on the west and Klaipeda on the east side of the route. The **x**'s correspond to the central lidar position in the hourly average.

The same process of co-location was carried out for the FINO2 data when SST information was recorded (2013–2017), where only one ASCAT grid cell enveloping the FINO2 position was used. This accounts for 2688 1 h complete collocations.

### 2.6. Atmospheric Stability Calculation

As meteorological measurements of atmospheric stability are uncommon in deep sea locations, we apply the bulk Richardson method from profile measurements [37] to obtain the stability correction for the logarithmic wind profile. For both the FINO2 met mast location and the moving ferry lidar, the wind speed at the lowest height $v_h$, the temperature at that height $T_h$, the difference in height from the sea-surface $z_h$, and the differences in the virtual potential temperatures at sea level $\Delta\Theta_v = \Theta_{v,h} - \Theta_{v,SST}$ were used to derive the dimensionless bulk Richardson number following [38]:

$$Ri_b = \frac{g}{\Theta_{v,h}} \frac{z_h \Delta\Theta_v}{v_h^2} \tag{1}$$

where $g$ is the acceleration due to gravity. The dimensionless stability parameter is thus as follows for stable and unstable cases:

$$\zeta = \begin{cases} \frac{10 R_{i_b}}{1 - 5 R_{i_b}}, & R_{i_b} > 0 \\ 10 R_{i_b}, & R_{i_b} \leq 0 \end{cases} \tag{2}$$

To extrapolate the profiles of the lidar and FINO2 down to the 10 m height, the logarithmic wind profile was used following [38]:

$$u(z) = \frac{u_*}{\kappa} \ln \left( \frac{z}{z_0} - \Psi_m(z/L) \right) \tag{3}$$

where $u_*$ is the frictional velocity, $z_0$ is the roughness length, $\kappa$ is the von Karman constant ($\kappa = 0.4$), and $\Psi_m(z/L)$ is the stability correction function introduced by [39] and [40]. The Obukhov length $L$ is obtained from the stability parameter $\zeta = z_h/L$, while $z_0$ and $u_*$ were calculated for each individual wind profile by using least-squares fitting of the lower portion of the wind profile where an interval is set to $1.5^{-5}{:}3^{-3}$, restricting the $z_0$ value based on the ocean surface roughness lengths in the work of [41]. The same stability classification scheme is used as in [42], which is shown in Table 1. A resultant extrapolated collocated profile can be seen in Figure 3. The full derivation of the stability correction can be found in Appendix A.

**Table 1.** Atmospheric stability classification scheme as suggested in [42].

| Stability Classification | Range |
| --- | --- |
| Very stable | $0.6 < \zeta < 2.0$ |
| Stable | $0.2 < \zeta < 0.6$ |
| Weakly stable | $0.02 < \zeta < 0.2$ |
| Neutral | $-0.02 < \zeta < 0.02$ |
| Weakly unstable | $-0.2 < \zeta < -0.02$ |
| Unstable | $-0.6 < \zeta < -0.2$ |
| Very Unstable | $-2.0 < \zeta < -0.6$ |

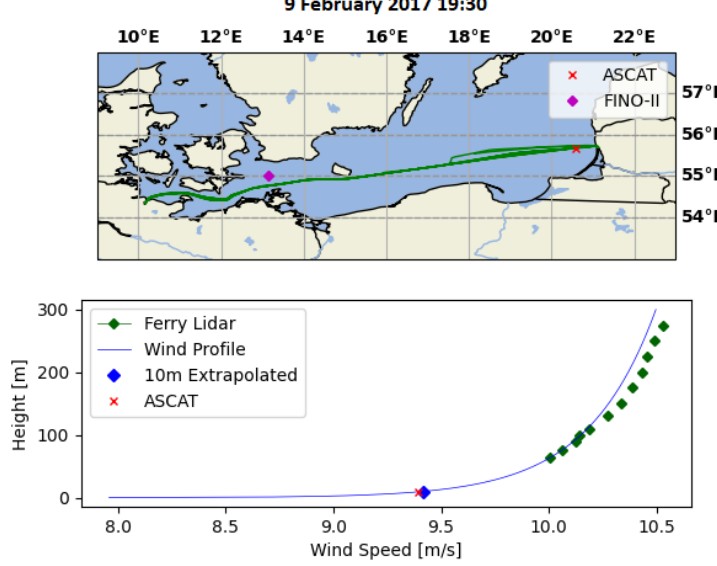

**Figure 3.** Wind profile extrapolation of co-located ASCAT and lidar measurements on 9 February 2017.

### 2.7. Mesoscale Model Simulations

The New European Wind Atlas uses the Weather Research and Forecasting (WRF) mesoscale model [15]. Mesoscale models cover a limited area and require boundary conditions from a global simulation system. To produce the best representation of the state of the atmosphere, data assimilation is used every 3 to 6 h with horizontal resolutions of tens of kilometers. The NEWA WRF modeling was carried out in a nested domain of high spatial resolution for a period of 4 years, where long-term wind statistics using the NCAR-NCEP (National Center for Atmospheric Research/National Centers for Environmental Prediction) reanalysis data were performed during 30 years to provide basis for a long-term adjustment of the results [15].

The NEWA WRF 30 min data was extracted in two different areas within the southern Baltic domain of the WRF production runs: the FINO2 location for the entire FINO2/ASCAT co-location period (N = 31,256) and the entire ferry lidar track at each 30 min measurement (N = 1181). Due to the limitation of the ASCAT co-locations, there were N = 2395 recorded WRF points at the FINO2 location. This is all summarized in Table 2.

**Table 2.** NEWA WRF 30 min collocation period and number of samples with each of the measurements.

|  | Period of Collocation | Number of Samples |
|---|---|---|
| FINO2 | 16 April 2013 to 30 November 2017 | 31,256 |
| ASCAT | 2 January 2007 to 29 December 2017 | 2395 |
| Ferry lidar | 13 February 2017 to 6 June 2017 | 1181 |

Similar to that presented in [12], the lidar/WRF co-located data were filtered for harbor effects where only data with a longitude coordinate between $10.4°$ and $20.0°$ were considered.

## 3. Results

### 3.1. Ferry Lidar vs. ASCAT

The co-located and extrapolated 10 m lidar data show very good coincidence with ASCAT 10 m wind speed and direction, where the entire evaluation period is shown in Figure 4. The scatter plots of the wind speeds from all data (left) and data filtered for near-neutral stability conditions (right, $-0.2 \leq \zeta \leq 0.2$) are shown in Figure 5. A large scatter and poor correlation is seen when looking at direct comparisons of the entire dataset, whereas filtering the data for near-neutral stability cases improves the correlation and reduces the root-mean-square-error (RMSE) to well within the acceptable OSI SAF product expected performance of ASCAT of 1.7 m s$^{-1}$ (see [31]). This is, however, reducing the already smaller dataset down to a fifth of the original observations with N = 23. The wind directions were not extrapolated down to the 10 m height so the comparisons with ASCAT are taken directly for the lowest possible height (i.e., 65 m). All of the biases were calculated with respect to that of the lidar, i.e., $(\overline{U}_{ASCAT} - \overline{U}_{lidar})/\overline{U}_{lidar}$. The overall statistical information of the wind speed and directions from the ferry lidar compared to ASCAT co-locations are summarized in Tables 3 and 4, respectively.

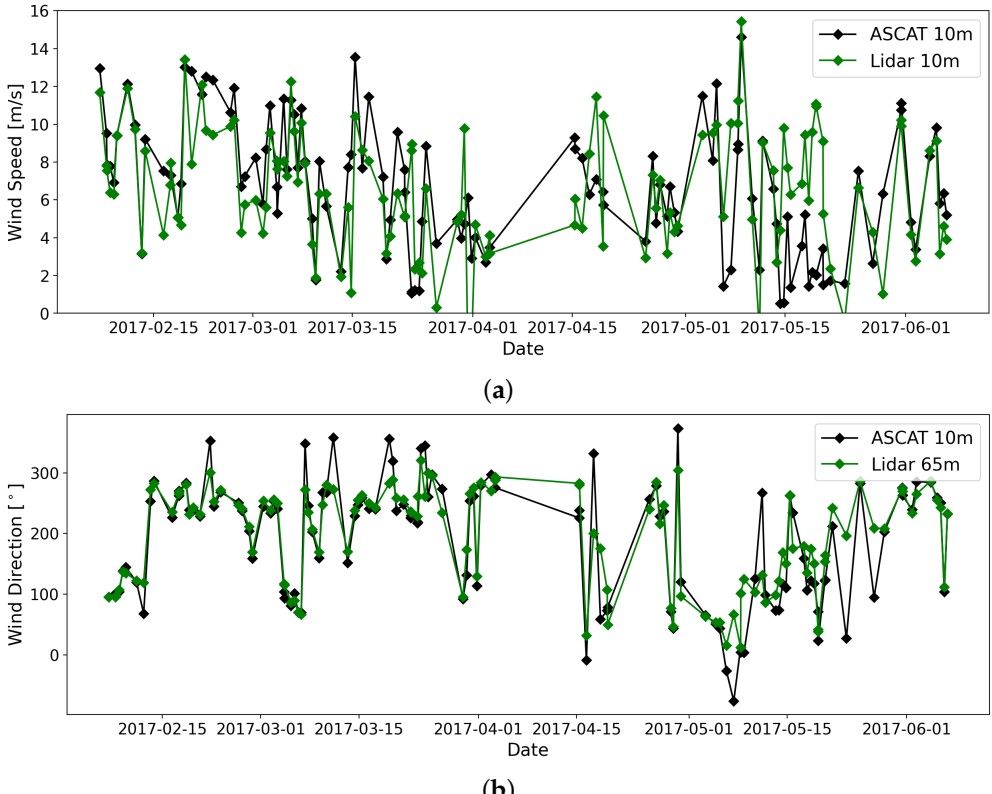

**Figure 4.** ASCAT-lidar co-location time series data: (**a**) 10 m wind speed extrapolation; (**b**) wind directions at different heights.

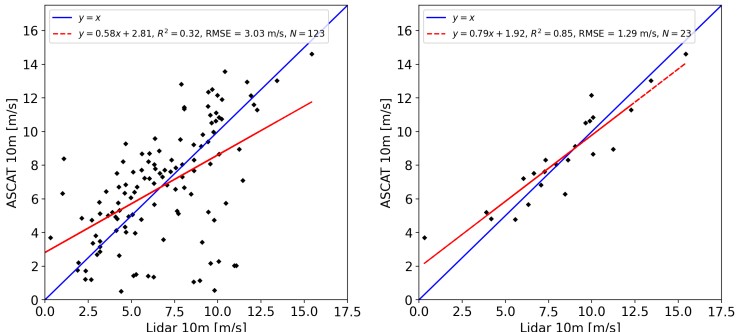

**Figure 5.** Scatter of extrapolated 10 m ferry lidar wind speed measurements with ASCAT 10 m. (**left**) All co-located extrapolated values (**right**) are filtered for near neutral stability ($-0.2 \leq \zeta \leq 0.2$). The blue line is $y = x$ and the red line is the resultant linear regression.

The wind direction comparisons in Table 4 have a higher overall error with those of ASCAT due to the fact that the data were compared at different heights. The wind directional data was not extrapolated such as the wind speed so the large error is mainly attributed to this expected veer in wind direction with height. Despite these large errors, the time series data in Figure 4 show a very similar behavior.

**Table 3.** Ferry lidar statistical characteristics (bias, RMSE, correlation) of wind speed with ASCAT, WRF, and FINO2.

| | Lidar Height (m) | Height (m) | $R^2$ | RMSE (m s$^{-1}$) | Bias (m s$^{-1}$) | N |
|---|---|---|---|---|---|---|
| Ferry lidar vs. ASCAT (collocated) | 10 | 10 | 0.85 | **1.29** | 0.02 | 23 |
| Ferry lidar vs. WRF (collocated) | 10 | 10 | 0.55 | 2.41 | 0.01 | 23 |
| | 65 | 50 | 0.76 | 1.90 | −0.01 | 3671 |
| | 75 | 75 | 0.76 | 1.93 | −0.02 | 3671 |
| | 100 | 100 | 0.77 | 2.04 | −0.00 | 3671 |
| | 200 | 200 | 0.79 | 2.27 | −0.02 | 3671 |
| | 250 | 250 | 0.80 | 2.32 | −0.03 | 3671 |
| Ferry lidar vs. FINO2 (30 km distance) | 65 | 62 | 0.87 | 1.71 | −0.11 | 134 |
| | 75 | 72 | 0.87 | 1.79 | −0.11 | 134 |
| | 90 | 92 | 0.87 | 1.81 | −0.07 | 134 |
| | 100 | 102 | 0.89 | 1.69 | −0.06 | 134 |

**Table 4.** Ferry lidar validation statistics (bias, RMSE, correlation) of wind direction with ASCAT, WRF, and FINO2.

| | Lidar Height (m) | Height (m) | $R^2$ | RMSE (°) | Bias (°) | N |
|---|---|---|---|---|---|---|
| Ferry lidar vs. ASCAT (collocated) | 65 | 10 | 0.83 | 34.8 | 0.06 | 119 |
| Ferry lidar vs. WRF (collocated) | 65 | 10 | 0.81 | 53.8 | 0.04 | 119 |
| | 65 | 50 | 0.79 | 26.3 | 0.01 | 3671 |
| | 75 | 75 | 0.78 | 25.6 | 0.01 | 3671 |
| | 100 | 100 | 0.76 | 25.4 | 0.01 | 3671 |
| | 200 | 200 | 0.73 | 28.1 | −0.01 | 3671 |
| | 250 | 250 | 0.75 | 27.2 | −0.00 | 3671 |
| Ferry lidar vs. FINO2 (30 km distance) | 75 | 71 | 0.86 | 32.8 | 0.04 | 134 |
| | 90 | 91 | 0.97 | 14.5 | 0.02 | 134 |

### 3.2. Ferry Lidar vs. FINO2

The overall statistical information for both the wind speed and direction of the ferry lidar with the other datasets are presented in Tables 3 and 4, respectively. The FINO2 dataset was directly compared to the ferry lidar at the closest available height, i.e., the 62 m wind speed at FINO2 was compared with 65 m from the ferry lidar, 72 m with 75 m, and 92 m with 90 m, up to 102 m and 100 m. Direct comparisons were performed at a maximum distance range of 30 km using 1 h measurements. Different timescales for direct comparisons were assessed (not shown), with the best results obtained when using 1 h averages. The 30 km range was chosen as it resulted in the highest number of collocated samples (N = 100) while only being 6 km away from the minimum distance (∼24 km). Different methods of comparisons were also studied (i.e., time delay, wind direction dependency, ship position filtering); however, the direct comparison at 1 h intervals yielded the best results.

The ferry lidar/FINO2 results consistently have a high correlation at all heights of $R^2 = 0.87$ or greater; nevertheless, due to the distance range of comparison with FINO2, we see a lower overall RMSE with that of the ferry lidar (1.70 m s$^{-1}$) than compared to the ASCAT–lidar comparison RMSE (1.29 m s$^{-1}$). Using the major wind directions at FINO2 during the day between 180° and 270°, the results yield the lower RMSE compared to all other sources as well as low bias and the best correlation. The wind directions at FINO2 change diurnally with a stronger westerly wind during the night. As the lidar passed by FINO2 during the day, the dominant wind directions of this time period were used.

### 3.3. ASCAT vs. FINO2

The scatter plots of the extrapolated FINO2 10 m wind speeds and ASCAT, both for all data and those filtered for near-neutral stabilities, are shown in Figure 6. When all available data are considered, independent of stability, there is a much stronger correlation between ASCAT and FINO2 than between ASCAT and the ferry lidar measurements. The RMSE is 1.51 m s$^{-1}$, significantly lower compared to 3.03 m s$^{-1}$ that was found for the ASCAT–ferry lidar comparison. When data were filtered for near-neutral atmospheric conditions, RMSE was reduced by 0.38 m s$^{-1}$ down to 1.13 m s$^{-1}$, and the correlation R$^2$ increased to 0.9, based on 923 samples. These values are comparable to what was reported under near-neutral cases between ASCAT and the ferry lidar comparisons, although these were based on a much smaller sample size.

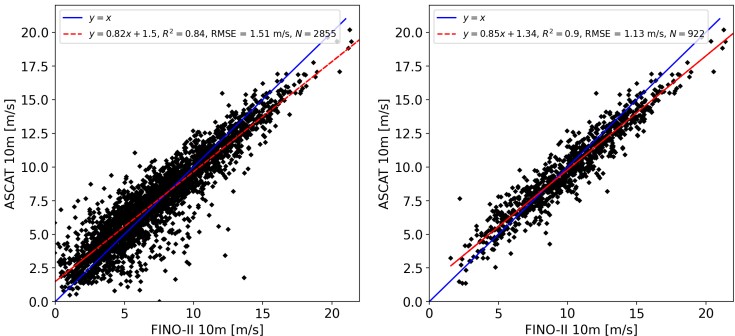

**Figure 6.** Scatter of extrapolated 10 m FINO2 wind speed measurements with ASCAT 10 m. (**left**) All co-located extrapolated values (**right**) filtered for near neutral stabilities ($-0.2 \leq \zeta \leq 0.2$). The blue line is $y = x$ and the red line is the resultant linear regression.

### 3.4. WRF

WRF has higher overall RMSE values and lower R$^2$ compared to the intercomparisons of the other measured datasets, all summarized in Table 5. FINO2 yielded the highest overall error at both heights which is consistent with [14,43]. This is seen in both the 30 min direct comparison and the 1 h comparison similar to that in [14]. ASCAT has the lowest overall error compared with the other direct comparisons; however, falls outside the expected performance of ASCAT (RMSE of 1.87 m s$^{-1}$).

**Table 5.** RMSE results of the NEWA WRF data compared to the other data sources for the wind speed.

|  | WRF Height (m) | Height (m) | R$^2$ | RMSE (m s$^{-1}$) | Bias (m s$^{-1}$) | N |
|---|---|---|---|---|---|---|
| WRF vs. ASCAT | 10 | 10 | 0.78 | 1.87 | 0.03 | 129 |
| WRF vs. FINO2 | 50 | 52 | 0.67 | 2.45 | 0.00 | 2867 |
|  | 100 | 102 | 0.70 | 2.61 | −0.03 | 2867 |
| WRF vs. Ferry Lidar | 50 | 65 | 0.76 | 1.90 | −0.01 | 3671 |
|  | 100 | 100 | 0.77 | 2.04 | −0.00 | 3671 |
|  | 200 | 200 | 0.79 | 2.27 | −0.02 | 3671 |
|  | 250 | 250 | 0.80 | 2.32 | −0.03 | 3671 |

Wind direction comparisons in Table 6 show better results than those for wind speed. ASCAT shows the smallest error, whereas the ferry lidar has a consistently higher source of error at all heights. The WRF–FINO2 comparison has a surprisingly high RMSE above 40° at both available heights on comparison. This is not as evident when looking at the time series of the WRF and ASCAT 10 m height wind directions alongside that of the 31 m FINO2 measurements in Figure 7.

**Table 6.** RMSE results of the NEWA WRF data compared to the other data sources for the wind direction.

| | WRF Height (m) | Height (m) | R$^2$ | RMSE (°) | Bias (°) | N |
|---|---|---|---|---|---|---|
| WRF vs. ASCAT | 10 | 10 | 0.95 | 17.6 | −0.01 | 129 |
| WRF vs. FINO2 | 50 | 51 | 0.58 | 41.8 | 0.02 | 2867 |
| | 100 | 91 | 0.58 | 42.6 | 0.02 | 2867 |
| WRF vs. Ferry Lidar | 50 | 65 | 0.79 | 26.3 | 0.01 | 3671 |
| | 100 | 100 | 0.76 | 25.4 | 0.01 | 3671 |
| | 200 | 200 | 0.73 | 28.1 | −0.01 | 3671 |
| | 250 | 250 | 0.75 | 27.2 | −0.00 | 3671 |

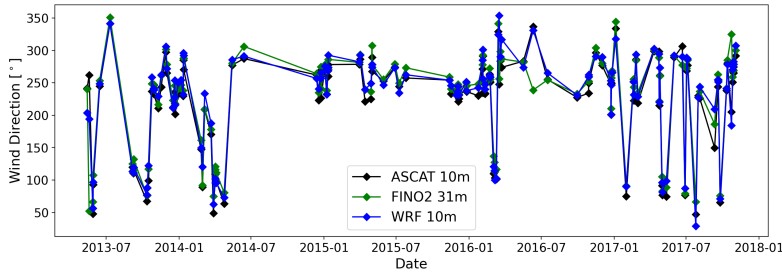

**Figure 7.** Wind direction timeseries of FINO2, ASCAT, and WRF at the FINO2 location from 2013 to 2017.

*3.5. Low-Level Jet Case Study*

The Baltic Sea is a semi-enclosed basin, meaning that the coastline–sea discontinuity has a strong influence on the wind conditions. During the spring months when the atmospheric stratification is, on average, stable, the temperature differences between land and sea have been shown to be 20 K or greater [24]. When this flow causes an air–sea temperature difference, an area of reduced turbulent mixing can persist for hundreds of kilometers. This increases the influence of inertial oscillations and thus increases the frequency of low-level jet (LLJ) occurrences, especially in coastal areas. Using the definition of [44], an LLJ is defined at the lowest maximum of the wind profile that is at least 2 m s$^{-1}$ and 25% faster than the next minimum.

Taking this into consideration, in around 30% of the 119 ASCAT–lidar collocations, we saw LLJ wind profiles such as in Figure 8. In this example, the discrepancy of the ASCAT data point and the extrapolated lidar measurements is around 5 m s$^{-1}$, which is a significant contributor to the large error seen in the original extrapolated data comparison. The majority of these LLJ events that affect the wind profile fitting process are filtered out when using the definition for near-neutral stratification. We see a larger overall occurrence of LLJs in the time period of the ferry lidar experiment mainly due to two reasons; the first being that, on average, the spring months have stable atmospheric stratification. The other contributor is the fact that a ship was docked at each port for 4 h during the night, where a larger portion of the stable atmospheric conditions occur during the day.

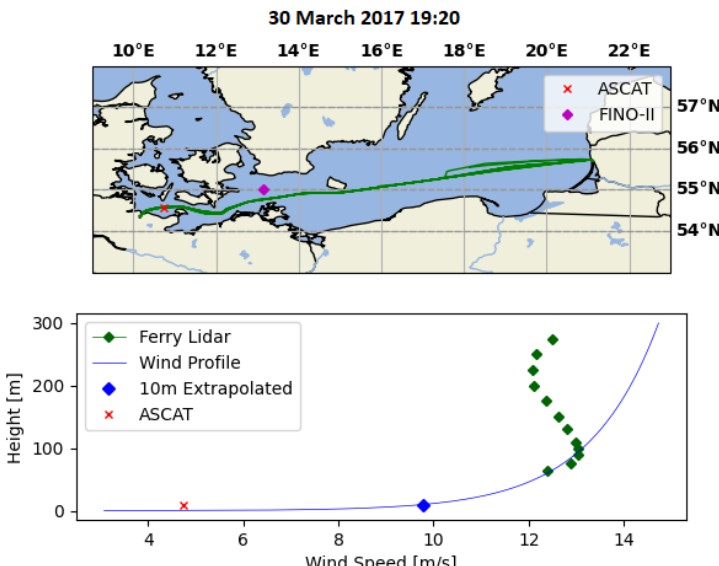

**Figure 8.** Wind profile extrapolation of co-located ASCAT and lidar measurements on 30 March 2017 with an LLJ.

## 4. Discussion

The present study has highlighted the complexity of intercomparing different types of wind speed and direction information. Each type of dataset, either from direct and indirect measurements or simulations, showed advantages and disadvantages. The FINO2 met mast represents the most accurate and direct measurement technique at a specific location and thus is lacking spatial representation. Furthermore, its distance to the ferry track rendered direct comparison with the lidar measurements more uncertain, and even more so, due to the spatial averaging of the lidar measurements themselves. It would be preferable to have a met mast located closer to the ferry track but this is not always possible, especially with the limited number of offshore met masts.

ASCAT was representative of much larger spatial scales with wide coverage over large parts of the study area; however, for the short campaign duration the number of overpasses available for direct comparison with the lidar measurements was limited along with the single reference height of 10 m. WRF covers spatial scales at multiple heights, but even the direct wind speed comparisons indicated that it was not able to reproduce the measured quantities. FINO2 shows very high error compared to WRF in both wind speed and direction, which is consistent with the results in both [43] and [14]. This may be due to the proximity of FINO2 to the Baltic II wind farm and the coast. However, we do not see this same level of error in the FINO2/lidar comparison at a larger distance scale and with the wind farm lying in-between the two sources with some of the lidar positions. Overall, WRF yielded the worst comparisons with all datasets at multiple heights.

One of the main challenges facing buoy-based FLS is the need for a reliable, autonomous, and robust setup that can withstand the rough offshore conditions. Ferry lidar systems offer a more reliable solution to this challenge with a more stable platform and a simpler means of installing and maintaining the lidar. This comes at the cost of sacrificing the flexibility that comes with buoys. Buoys can be relocated with relative ease and it is much easier and faster to obtain permissions than other meteorological measurement techniques. Ferry lidars offer the possibility to be installed on already-existent ferry routes, especially ones that correspond to areas of interest. Ship-based lidars are still in their infancy and need a more robust validation process for more widespread use, especially to become a tool for wind resource assessment.

Unlike the buoy-based lidar counterpart, ship-based lidars do not have the same scrutiny or recommended practices already established. The IEA Wind Task 32 [8] gives

detailed guidance into the recommended practices of FLS. This includes verifying the FLS with a reference no more than 500 m away, using 10 min average wind speeds and recommending a six-month campaign duration for adequate data coverage and met–ocean conditions. All of these conditions are not met within this campaign, with the exception of using the 10 min mean wind speed. However, with the spatial coverage the ferry envelopes, using 10 min averages does not cover the WRF or ASCAT spatial domains and covers a larger area when compared to FINO2. Zentek et al. [13] use radiosondes in conjunction with the ship lidar as means of verification but this results in instantaneous comparisons. In the validation campaign from Bremenhaven to Helgoland (see [29]) from 2 June 2016 to 22 August 2016, the ferry lidar was validated against met mast data in the two ports at distances of 1.14 km and 500 m, respectively. In both of these cases, the ferry was close to, or within range of, the recommended practices; however, it was approaching port and therefore not at the top speed, such as in open water. This is one of the main challenges in ferry-based lidar measurements; it is very difficult to study/validate the corrected wind speed data in open water or at top speeds. This means that ferry lidar systems cannot follow the same recommended practices of FLS and are more difficult to achieve a "pre-commercial" maturity level.

A future outlook is the use of machine learning to extrapolate the satellite data to lidar heights for a direct comparison. This is foreseen through training models on low-level atmospheric data and predicting wind speeds at multiple heights similar to that of [45]. Having more lidar systems mounted on existing ferry routes would help bridge the gap of data scarcity offshore and provide more datasets to train extrapolation models. WRF provides similar spatial information desired as well as values at multiple heights, but has been shown to produce the largest overall error where the extrapolated satellite measurements could bridge this gap. In [12], where the ferry lidar experiment was first introduced, the RMSE results compared to their WRF simulations had a similar error to those of the NEWA WRF dataset used in this work. We record an RMSE in wind speed of 2.04 m s$^{-1}$ for all data, whereas that study recorded a value of 1.91 m s$^{-1}$ using model setup for initial verification of ferry lidar results before this experiment. This is not a significant difference and we see the same results in the wind directions; our results show an RMSE of 25.4°, whereas this group shows 29.2°. Even using a WRF model parameterized to the Baltic Sea for the period when the ferry-lidar experiment was conducted, the error was still greater than that what is found using FINO2 and ASCAT presented in this work.

ASCAT has shown to be an invaluable source of data offshore, with relatively high frequency of measurements compared to other satellite sources as well as lower overall standard deviations and biases. Even with the difficulty to directly compare the results with those of hub height levels, the results compared to the extrapolated wind profiles proved significant. All ASCAT comparison results fall within the acceptable OSI SAF product statistical range with the exception that the lidar extrapolated 10 m wind speeds when all atmospheric conditions were included. This yields an error of 3.0 m s$^{-1}$, twice that of the FINO2 equivalent comparison. This discrepancy is mainly due to the period of measurement and measurement heights. From February to June, the atmosphere is, on average, stably stratified; as was found from calculations in this study, stable profiles appeared over 50% of the time. The lowest lidar measuring height from sea level was at 65 m, which means we were not able to properly fit to the lower portion of the wind profile during stable conditions. As turbulent fluxes decrease in magnitude and near-surface winds decouple from the wind aloft [46,47], the surface layer is not represented at a height of 65 m, and thus is not properly representing the wind at 10 m when extrapolated downwards.

The lower surface layer experiencing stable stratification and the high frequency of low-level jets in the Baltic Sea [25] cause the wind profiles in some cases to invert at lower heights where the profiles are being fit, which is not accounted for in the logarithmic wind profile. Thus, filtering for only near-neutral atmospheric stratification reduces errors from 3.0 m s$^{-1}$ to a level similar to what was found for the FINO2 comparison; however, it results in a much smaller dataset size (N = 23) that may be too small to deem credible or even to

take into consideration when comparing to the other measured quantities. The amount of scatter, represented in Figure 6, between ASCAT and FINO2 is smaller than what was found between ASCAT and the lidar measurements, although the former comparison extends over 4 years and has many more collocated pairs. This is mainly attributed to the lower measurement heights of FINO2, i.e., lower portion of the wind profile consistently representing the surface layer.

## 5. Conclusions

This study presents intercomparisons of the NEWA ferry lidar campaign in the southern Baltic Sea, from February to June 2017, with wind retrievals from ASCAT, in situ measurements from the FINO2 mast, and simulated winds using the WRF mesoscale model. Furthermore, intercomparisons between FINO2, WRF, and ASCAT were performed from 2013 to 2017. Challenges in intercomparing with the moving lidar measurements were associated with collocation practices, e.g., comparing spatially distant measurements, spatially averaged measurements, or measurements at varying heights. This resulted in varying RMSE values for the wind speed between the ferry lidar measurements and other sources, with an overall lowest RMSE of 1.29 m s$^{-1}$ between ASCAT and the ferry lidar at a height of 10 m.

FINO2, as the only true in situ source of wind measurements, could be considered the reference for the comparisons, yet its significant distance from the ferry lidar track was assumed to be the cause of the larger-than-expected errors in wind speed around 1.70 m s$^{-1}$. Nonetheless, the lowest error in wind directions was found for the FINO2 comparisons with the ferry lidar. The highest errors for the WRF-simulated winds were found when compared to the FINO2 measurements, consistent with the previous studies in this area.

Overall, the ferry lidar is a valuable tool using lidar technology to cover wind-energy-relevant scales (temporal and spatial). It is, however, difficult to validate and has yet to receive recommended practices. Further study at top speeds is recommended, using a concurrence of radiosondes, stabilized lidar devices, or satellite winds extrapolated with machine learning.

**Author Contributions:** D.H. prepared the original draft, as well as acquired, developed, and performed the data analysis and produced the results. C.B.H. and I.K. contributed in numerous discussions, provided suggestions, and supported the interpretation of the results. All authors reviewed and edited the manuscript until it reached the final stage. All authors have read and agreed to the published version of the manuscript.

**Funding:** This project has received funding from the European Union's Horizon 2020 research and innovation programme under the Marie Sklodowska-Curie grant agreement number 860879.

**Institutional Review Board Statement:** Not applicable.

**Informed Consent Statement:** Not applicable.

**Data Availability Statement:** The New European Wind Atlas is published at https://map.neweuropeanwindatlas.eu/ (last access: 15 February 2021; NEWA, 2021). The OSTIA dataset can be obtained from http://marine.copernicus.eu/ (last access: 25 February 2022; Copernicus marine service, 2022). The ASCAT data was taken from https://marine.copernicus.eu (last access: 14 March 2022) The FINO2 data can be obtained from http://fino.bsh.de (last access: 12 November 2021, FINO2, 2021).

**Acknowledgments:** We acknowledge the NEWA consortium for providing access to the New European Wind Atlas. We acknowledge Fraunhofer IWES and Julia Gottschall for providing the ferry lidar data set.

**Conflicts of Interest:** The authors declare no conflicts of interest.

## Appendix A. Stability Correction through Bulk Richardson Number

The following derivation is adapted from [38,48,49].

Assuming a polytropic atmosphere, the air temperature gradient, $\gamma$ is

$$\gamma = \frac{T_{SST} - T_h}{z_h}$$

where $h$ is the height used, in the case of the FINO2, this is $h = 30$ m, and temperatures are in Kelvin.

The air pressure at sea level is estimated to be

$$p_0 = p_h \left( \frac{T_{SST} - \gamma z_h}{T_{SST}} \right)^{\frac{-g}{\gamma R_d}}$$

where the acceleration due to gravity is $g = 9.81\,\text{ms}^{-2}$ and the specific gas constant of water vapor is $R_d = 287\,\text{JK}^{-1}\text{kg}^{-1}$. The saturation vapor pressure is dependent on the temperature through the Magnus equation:

$$e_s(T) = 100 \cdot 6.1 \cdot 10^{\frac{7.45(T_h - 273.15)}{T_h - 38.15}}$$

Here, the partial pressure of water vapor in the air is dependent on the relative humidity, RH:

$$e = \text{RH} \cdot e_s / 100$$

where the mixing ratio is

$$r_v = \frac{R_d}{R_v} \cdot \left( \frac{e}{p_h - e} \right)$$

where the specific gas constant of water vapor is $R_v = 461\,\text{JK}^{-1}\text{kg}^{-1}$. The potential temperature, $\Theta$, is calculated to remove the temperature variation caused by changes in pressure as

$$\Theta = T_h \left( \frac{p_0}{p_h} \right)^{\kappa_p}$$

with $\kappa_p = 0.286$ being the Poisson constant assuming dry air. We can write the specific humidity as

$$q = \frac{r_v}{1 + r_v}$$

Now, the virtual potential temperature is calculated to account for density in buoyancy calculations and in turbulence transport which includes vertical air movement.

$$\Theta_v = \Theta \cdot (1.0 + 0.61q)$$

Finally, the bulk Richardson number can be calculated as

$$\text{Ri}_b = \frac{g \Delta \Theta_v z_h}{\Theta_v u_h^2} \tag{A1}$$

The bulk Richardson number is a dimensionless ratio of the consumption of turbulence and the production by wind shear of turbulence. It is then further used to show the dynamic stability through estimating the dimensionless stability parameter:

$$\zeta = \begin{cases} \frac{10 R_{i_b}}{1 - 5 R_{i_b}}, & R_{i_b} > 0 \\ 10 R_{i_b}, & R_{i_b} \leq 0 \end{cases} \tag{A2}$$

In unstable stratifications ($\zeta < 0$), the stability correction in M-O theory can then be estimated by

$$\Psi_m = 2\ln\left( \frac{1+x}{2} \right) + \ln\left( \frac{1+x^2}{2} \right) - 2\arctan(x) + \frac{\pi}{2} \tag{A3}$$

with $x = (1 - b\,z/L_*)^{1/4}$ where $z/L_* = \zeta$ (from [39,40]).

Stable stratifications are broken down into two cases:

$$\Psi = \begin{cases} -C\zeta, & 0 \le \zeta \le 0.5 \\ A\zeta + B(\zeta - (C/D))\exp(-D\zeta) + B(C/D), & 0.5 \le \zeta \le 7.0 \end{cases} \tag{A4}$$

where $A = 1$, $B = 2/3$, $C = 5$, and $D = 0.35$ (from [50–52]; summarized in [38]).

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
