# Peer review of "Comparing Offshore Ferry Lidar Measurements in the Southern Baltic Sea with ASCAT, FINO2 and WRF"

_remotesensing, doi:10.3390/rs14061427_

Round 1

Reviewer 1 Report

Overall Comments:

In this manuscript, the authors introduced the comparison between the Ferry Lidar Experiment wind data and those measured by ASCAT, FINO2, and WRF. The study evaluated the intensity and limitations of the above Ferry Lidar experiments using the available datasets of ASCAT and FINO2 in the offshore area of the southern Baltic Sea from February to June 2017. Inter-comparisons between these datasets also were conducted from 2013 to 2017.

The results demonstrated that the wind speed RMSE values measured by the ferry lidar are different from those from other sources. This study combines the advantages and disadvantages of various data sources and obtains the best estimation of wind power generation potential. The structure of this manuscript is logical, which has certain practical significance.

However, the manuscript still has several issues. Below are specific comments/questions that need to be addressed before publication.

Specific/Detailed Questions:

  1. The authors could add further details (i.e.: a small paragraph) about the earlier studies that other comparisons of the available datasets in this field.
  2. In section 3.2, the number of collocated samples in Ferry Lidar vs. ASCAT is only 23, based on this quantity, is it credible to obtain the following comparison results? Also please add the description of Figure 5 in your main text.
  3. The discussion could be more developed and different types of discussions could be divided into sections. (g. The reasons for FINO2 error speed and direction could be more specific)
  4. Figure 7 doesn't have the unit in the colorbar. Some figures’ axis labels/units are not clear and bold enough (e.g. Figure 4, 5 and 7).

Finally, there are some errors in the text that should be corrected. As an example:

  1. L330: that ship ==> that the ship
  2. L343: a wide coverage ==> a wide coverage
  3. L361: corresponds ==> correspond
  4. L393: in the the two ports ==> in the the two ports
  5. L422: is smaller that what was==> is smaller than what was

Reviewer 2 Report

Combining various measurement techniques with different resolution and accuracy profiles that also appear robust under different experimental conditions is essential for an improvement of sea surface characteristics. Based on a series of original experimental measurements and several available external data sources, the authors have shown explicitly that the lidar data provides (A) rather accurate and (B) complementary information on the wind fields over the sea surface. The paper reports original experimental data, extended simulations / reconstructions and statistical analysis and apparently deserves publication.

In the linear regression models in Figs. 5, 6 the slopes are all below one, indicating that there is a systematic measurement bias. Indeed, it could be obviously corrected, assuming that the slope bias is representative, although the latter may be dependent on various local conditions, and is likely difficult to prove explicitly based on the measurements from a single route. However, except for Fig. 5A, it looks like linear regression without an intercept term, or even just y=x would effect the model accuracy only moderately. In turn, in addition to the R^2 for the best linear model with intercept, one could also provide alternative R^2 values for the linear models without intercept, as well as for the y=x model, respectively (for recent examples see, e.g., https://journals.plos.org/plosone/article?id=10.1371/journal.pone.0193267; https://www.sciencedirect.com/science/article/pii/S0378375810000194; https://www.tandfonline.com/doi/abs/10.1080/03610926.2010.535631).

As an outlook, it would be interesting if the authors provide some further discussion on the issue (B), in particular, how they see potential scenarios of combining measurement sources, based on the difference in the methodological accuracy, performance and corresponding error patterns.

Although the paper is generally very well written, there appear a few technical issues, that could be easily corrected during a final proof reading. For example, line 34: “… has been … have been …”, one of them is likely redundant.
